# Nesterov Accelerated Gradient and Scale Invariance for Adversarial Attacks

**Jiadong Lin & Chuanbiao Song & Kun He** [*]
School of Computer Science and Technology
Huazhong University of Science and Technology
Wuhan, 430074, China
{jdlin,cbsong,brooklet60}@hust.edu.cn

**Liwei Wang**
School of Electronics Engineering
and Computer Sciences, Peking University
Peking, China
wanglw@cis.pku.edu.cn

**John E. Hopcroft**
Department of Computer Science
Cornell University, NY 14853, USA
jeh@cs.cornell.edu

## Abstract

Deep learning models are vulnerable to adversarial examples crafted by applying human-imperceptible perturbations on benign inputs. However, under the black-box setting, most existing adversaries often have a poor transferability to attack other defense models. In this work, from the perspective of regarding the adversarial example generation as an optimization process, we propose two new methods to improve the transferability of adversarial examples, namely Nesterov Iterative Fast Gradient Sign Method (NI-FGSM) and Scale-Invariant attack Method (SIM). NI-FGSM aims to adapt Nesterov accelerated gradient into the iterative attacks so as to effectively look ahead and improve the transferability of adversarial examples. While SIM is based on our discovery on the scale-invariant property of deep learning models, for which we leverage to optimize the adversarial perturbations over the scale copies of the input images so as to avoid "overfitting" on the white-box model being attacked and generate more transferable adversarial examples. NI-FGSM and SIM can be naturally integrated to build a robust gradient-based attack to generate more transferable adversarial examples against the defense models. Empirical results on ImageNet dataset demonstrate that our attack methods exhibit higher transferability and achieve higher attack success rates than state-of-the-art gradient-based attacks.

## 1 Introduction

Deep learning models have been shown to be vulnerable to adversarial examples (Goodfellow et al., 2014; Szegedy et al., 2014), which are generated by applying human-imperceptible perturbations on benign input to result in the misclassification. In addition, adversarial examples have an intriguing property of transferability, where adversarial examples crafted by the current model can also fool other unknown models. As adversarial examples can help identify the robustness of models (Arnab et al., 2018), as well as improve the robustness of models by adversarial training (Goodfellow et al., 2014), learning how to generate adversarial examples with high transferability is important and has gained increasing attentions in the literature (Liu et al., 2016; Dong et al., 2018; Xie et al., 2019; Dong et al., 2019; Wang et al., 2019).

Several gradient-based attacks have been proposed to generate adversarial examples, such as one-step attacks (Goodfellow et al., 2014) and iterative attacks (Kurakin et al., 2016; Dong et al., 2018). Under the white-box setting, with the knowledge of the current model, existing attacks can achieve high success rates. However, they often exhibit low success rates under the black-box setting, especially for models with defense mechanism, such as adversarial training (Madry et al., 2018; Song

---

[*]Corresponding author.

et al., 2019) and input modification (Liao et al., 2018; Xie et al., 2018). Under the black-box setting, most existing attacks fail to generate robust adversarial examples against defense models.

In this work, by regarding the adversarial example generation process as an optimization process, we propose two new methods to improve the transferability of adversarial examples: Nesterov Iterative Fast Gradient Sign Method (NI-FGSM) and Scale-Invariant attack Method (SIM).

- Inspired by the fact that Nesterov accelerated gradient (Nesterov, 1983) is superior to momentum for conventionally optimization (Sutskever et al., 2013), we adapt Nesterov accelerated gradient into the iterative gradient-based attack, so as to effectively look ahead and improve the transferability of adversarial examples. We expect that NI-FGSM could replace the momentum iterative gradient-based method (Dong et al., 2018) in the gradient accumulating portion and yield higher performance.
- Besides, we discover that deep learning models have the *scale-invariant* property, and propose a Scale-Invariant attack Method (SIM) to improve the transferability of adversarial examples by optimizing the adversarial perturbations over the scale copies of the input images. SIM can avoid "overfitting" on the white-box model being attacked and generate more transferable adversarial examples against other black-box models.
- We found that combining our NI-FGSM and SIM with existing gradient-based attack methods (e.g., diverse input method (Xie et al., 2019)) can further boost the attack success rates of adversarial examples.

Extensive experiments on the ImageNet dataset (Russakovsky et al., 2015) show that our methods attack both normally trained models and adversarially trained models with higher attack success rates than existing baseline attacks. Our best attack method, SI-NI-TI-DIM (Scale-Invariant Nesterov Iterative FGSM integrated with translation-invariant diverse input method), reaches an average success rate of 93.5% against adversarially trained models under the black-box setting. For further demonstration, we evaluate our methods by attacking the latest robust defense methods (Liao et al., 2018; Xie et al., 2018; Liu et al., 2019; Jia et al., 2019; Cohen et al., 2019). The results show that our attack methods can generate adversarial examples with higher transferability than state-of-the-art gradient-based attacks.

## 2 Preliminary

### 2.1 Notation

Let $x$ and $y^{true}$ be a benign image and the corresponding true label, respectively. Let $J(x, y^{true})$ be the loss function of the classifier (e.g. the cross-entropy loss). Let $x^{adv}$ be the adversarial example of the benign image $x$. The goal of the non-targeted adversaries is to search an adversarial example $x^{adv}$ to maximize the loss $J(x^{adv}, y^{true})$ in the $\ell_p$ norm bounded perturbations. To align with previous works, we focus on $p = \infty$ in this work to measure the distortion between $x^{adv}$ and $x$. That is $\left\| x^{adv} - x \right\|_\infty \leq \epsilon$, where $\epsilon$ is the magnitude of adversarial perturbations.

### 2.2 Attack Methods

Several attack methods have been proposed to generate adversarial examples. Here we provide a brief introduction.

**Fast Gradient Sign Method (FGSM).** FGSM (Goodfellow et al., 2014) generates an adversarial example $x^{adv}$ by maximizing the loss function $J(x^{adv}, y^{true})$ with one-step update as:

$$x^{adv} = x + \epsilon \cdot \text{sign}(\nabla_x J(x, y^{true})), \tag{1}$$

where $\text{sign}(\cdot)$ function restricts the perturbation in the $L_\infty$ norm bound.

**Iterative Fast Gradient Sign Method (I-FGSM).** Kurakin et al. (2016) extend FGSM to an iterative version by applying FGSM with a small step size $\alpha$:

$$x_0 = x, \quad x_{t+1}^{adv} = \text{Clip}_x^\epsilon \{ x_t^{adv} + \alpha \cdot \text{sign}(\nabla_x J(x_t^{adv}, y^{true})) \}, \tag{2}$$

where $\text{Clip}_x^\epsilon(\cdot)$ function restricts generated adversarial examples to be within the $\epsilon$-ball of $x$.

**Projected Gradient Descent (PGD).** PGD attack (Madry et al., 2018) is a strong iterative variant of FGSM. It consists of a random start within the allowed norm ball and then follows by running several iterations of I-FGSM to generate adversarial examples.

**Momentum Iterative Fast Gradient Sign Method (MI-FGSM).** Dong et al. (2018) integrate momentum into the iterative attack and lead to a higher transferability for adversarial examples. Their update procedure is formalized as follows:

$$g_{t+1} = \mu \cdot g_t + \frac{\nabla_x J(x_t^{adv}, y^{true})}{\left\| \nabla_x J(x_t^{adv}, y^{true}) \right\|_1},$$
$$x_{t+1}^{adv} = \text{Clip}_x^\epsilon \{ x_t^{adv} + \alpha \cdot \text{sign}(g_{t+1}) \}, \tag{3}$$

where $g_t$ is the accumulated gradient at iteration $t$, and $\mu$ is the decay factor of $g_t$.

**Diverse Input Method (DIM).** Xie et al. (2019) optimize the adversarial perturbations over the diverse transformation of the input image at each iteration. The transformations include the random resizing and the random padding. DIM can be naturally integrated into other gradient-based attacks to further improve the transferability of adversarial examples.

**Translation-Invariant Method (TIM).** Instead of optimizing the adversarial perturbations on a single image, Dong et al. (2019) use a set of translated images to optimize the adversarial perturbations. They further develop an efficient algorithm to calculate the gradients by convolving the gradient at untranslated images with a kernel matrix. TIM can also be naturally integrated with other gradient-based attack methods. The combination of TIM and DIM, namely TI-DIM, is the current strongest black-box attack method.

**Carlini & Wagner attack (C&W).** C&W attack (Carlini & Wagner, 2017) is an optimization-based method which directly optimizes the distance between the benign examples and the adversarial examples by solving:

$$\underset{x^{adv}}{\arg\min} \ \left\| x^{adv} - x \right\|_p - c \cdot J(x^{adv}, y^{true}). \tag{4}$$

It is a powerful method to find adversarial examples while minimizing perturbations for white-box attacks, but it lacks the transferability for black-box attacks.

## 2.3 DEFENSE METHODS

Various defense methods have been proposed to against adversarial examples, which can fall into the following two categories.

**Adversarial Training.** One popular and promising defense method is *adversarial training* (Goodfellow et al., 2014; Szegedy et al., 2014; Zhai et al., 2019; Song et al., 2020), which augments the training data by the adversarial examples in the training process. Madry et al. (2018) develop a successful adversarial training method, which leverages the projected gradient descent (PGD) attack to generate adversarial examples. However, this method is difficult to scale to large-scale datasets (Kurakin et al., 2017). Tramèr et al. (2018) propose *ensemble adversarial training* by augmenting the training data with perturbations transferred from various models , so as to further improve the robustness against the black-box attacks. Currently, adversarial training is still one of the best techniques to defend against adversarial attacks.

**Input Modification.** The second category of defense methods aims to mitigate the effects of adversarial perturbations by modifying the input data. Guo et al. (2018) discover that there exists a range of image transformations, which have the potential to remove adversarial perturbations while preserving the visual information of the images. Xie et al. (2018) mitigate the adversarial effects through random transformations. Liao et al. (2018) propose high-level representation guided denoiser to purify the adversarial examples. Liu et al. (2019) propose a JPEG-based defensive compression framework to rectify adversarial examples without impacting classification accuracy on benign data. Jia et al. (2019) leverage an end-to-end image compression model to defend adversarial examples. Although these defense methods perform well in practice, they can not tell whether the model is truly robust to adversarial perturbations. Cohen et al. (2019) use randomized smoothing to obtain an ImageNet classifier with certified adversarial robustness.

## 3 METHODOLOGY

### 3.1 MOTIVATION

Similar with the process of training neural networks, the process of generating adversarial examples can also be viewed as an optimization problem. In the optimizing phase, the white-box model being attacked to optimize the adversarial examples can be viewed as the training data on the training process. And the adversarial examples can be viewed as the training parameters of the model. Then in the testing phase, the black-box models to evaluate the adversarial examples can be viewed as the testing data of the model.

From the perspective of the optimization, the transferability of the adversarial examples is similar with the generalization ability of the trained models (Dong et al., 2018). Thus, we can migrate the methods used to improve the generalization of models to the generation of adversarial examples, so as to improving the transferability of adversarial examples.

Many methods have been proposed to improve the generalization ability of the deep learning models, which can be split to two aspects: (1) better optimization algorithm, such as Adam optimizer(Kingma & Ba, 2014); (2) data augmentation (Simonyan & Zisserman, 2014). Correspondingly, the methods to improve the transferability of adversarial examples can also be split to two aspects: (1) better optimization algorithm, such as MI-FGSM, which applies the idea of momentum; (2) model augmentation (i.e., ensemble attack on multiple models), such as the work of Dong et al. (2018), which considers to attack multiple models simultaneously. Based on above analysis, we aim to improve the transferability of adversarial examples by applying the idea of Nesterov accelerated gradient for *optimization* and using a set of scaled images to achieve *model augmentation*.

### 3.2 NESTEROV ITERATIVE FAST GRADIENT SIGN METHOD

Nesterov Accelerated Gradient (NAG) (Nesterov, 1983) is a slight variation of normal gradient descent, which can speed up the training process and improve the convergence significantly. NAG can be viewed as an improved momentum method, which can be expressed as:

$$
\begin{aligned}
v_{t+1} &= \mu \cdot v_t + \nabla_{\theta_t} J(\theta_t - \alpha \cdot \mu \cdot v_t), \\
\theta_{t+1} &= \theta_t - \alpha \cdot v_{t+1}.
\end{aligned}
\tag{5}
$$

Typical gradient-based iterative attacks (e.g., I-FGSM) greedily perturb the images in the direction of the sign of the gradient at each iteration, which usually falls into poor local maxima, and shows weak transferability than single-step attacks (e.g., FGSM). Dong et al. (2018) show that adopting momentum (Polyak, 1964) into attacks can stabilize the update directions, which helps to escape from poor local maxima and improve the transferability. Compared to momentum, beyond stabilize the update directions, the anticipatory update of NAG gives previous accumulated gradient a correction that helps to effectively look ahead. Such looking ahead property of NAG can help us escape from poor local maxima easier and faster, resulting in the improvement on transferability.

We integrate NAG into the iterative gradient-based attack to leverage the looking ahead property of NAG and build a robust adversarial attack, which we refer to as NI-FGSM (Nesterov Iterative Fast Gradient Sign Method). Specifically, we make a jump in the direction of previous accumulated gradients before computing the gradients in each iteration. Start with $g_0 = 0$, the update procedure of NI-FGSM can be formalized as follows:

$$
x_t^{nes} = x_t^{adv} + \alpha \cdot \mu \cdot g_t,
\tag{6}
$$

$$
g_{t+1} = \mu \cdot g_t + \frac{\nabla_x J(x_t^{nes}, y^{true})}{\|\nabla_x J(x_t^{nes}, y^{true})\|_1},
\tag{7}
$$

$$
x_{t+1}^{adv} = \text{Clip}_x^\epsilon \{x_t^{adv} + \alpha \cdot \text{sign}(g_{t+1})\},
\tag{8}
$$

where $g_t$ denotes the accumulated gradients at the iteration $t$, and $\mu$ denotes the decay factor of $g_t$.

### 3.3 SCALE-INVARIANT ATTACK METHOD

Besides considering a better optimization algorithm for the adversaries, we can also improve the transferability of adversarial examples by *model augmentation*. We first introduce a formal definition of loss-preserving transformation and model augmentation as follows.

**Definition 1** *Loss-preserving Transformation. Given an input $x$ with its ground-truth label $y^{true}$ and a classifier $f(x) : x \in \mathcal{X} \to y \in \mathcal{Y}$ with the cross-entropy loss $J(x, y)$, if there exists an input transformation $\mathcal{T}(\cdot)$ that satisfies $J(\mathcal{T}(x), y^{true}) \approx J(x, y^{true})$ for any $x \in \mathcal{X}$, we say $\mathcal{T}(\cdot)$ is a loss-preserving transformation.*

**Definition 2** *Model Augmentation. Given an input $x$ with its ground-truth label $y^{true}$ and a model $f(x) : x \in \mathcal{X} \to y \in \mathcal{Y}$ with the cross-entropy loss $J(x, y)$, if there exists a loss-preserving transformation $\mathcal{T}(\cdot)$, then we derive a new model by $f'(x) = f(\mathcal{T}(x))$ from the original model $f$. we define such derivation of models as model augmentation.*

Intuitively, similar to the generalization of models that can be improved by feeding more training data, the transferability of adversarial examples can be improved by attacking more models simultaneously. Dong et al. (2018) enhance the gradient-based attack by attacking an ensemble of models. However, their approach requires training a set of different models to attack, which has a large computational cost. Instead, in this work, we derive an ensemble of models from the original model by *model augmentation*, which is a simple way of obtaining multiple models via the loss-preserving transformation.

To get the loss-preserving transformation, we discover that deep neural networks might have the scale-invariant property, besides the translation invariance. Specifically, the loss values are similar for the original and the scaled images on the same model, which is empirically validated in Section 4.2. Thus, the scale transformation can be served as a model augmentation method. Driven by the above analysis, we propose a Scale-Invariant attack Method (SIM), which optimizes the adversarial perturbations over the scale copies of the input image:

$$\underset{x^{adv}}{\arg\max} \frac{1}{m} \sum_{i=0}^{m} J(S_i(x^{adv}), y^{true}), \tag{9}$$
$$\text{s.t. } \left\| x^{adv} - x \right\|_{\infty} \leq \epsilon,$$

where $S_i(x) = x/2^i$ denotes the scale copy of the input image $x$ with the scale factor $1/2^i$, and $m$ denotes the number of the scale copies. With SIM, instead of training a set of models to attack, we can effectively achieve ensemble attacks on multiple models by model augmentation. More importantly, it can help avoid "overfitting" on the white-box model being attacked and generate more transferable adversarial examples.

## 3.4 ATTACK ALGORITHM

For the gradient processing of crafting adversarial examples, NI-FGSM introduces a better optimization algorithm to stabilize and correct the update directions at each iteration. For the ensemble attack of crafting adversarial examples, SIM introduces *model augmentation* to derive multiple models to attack from a single model. Thus, NI-FGSM and SIM can be naturally combined to build a stronger attack, which we refer to as SI-NI-FGSM (Scale-Invariant Nesterov Iterative Fast Gradient Sign Method). The algorithm of SI-NI-FGSM attack is summarized in Algorithm 1.

In addition, SI-NI-FGSM can be integrated with DIM (Diverse Input Method), TIM (Translation-Invariant Method) and TI-DIM (Translation-Invariant with Diverse Input Method) as SI-NI-DIM, SI-NI-TIM and SI-NI-TI-DIM, respectively, to further boost the transferability of adversarial examples. The detailed algorithms for these attack methods are provided in Appendix A.

## 4 EXPERIMENTAL RESULTS

In this section, we provide experimental evidence on the advantage of the proposed methods. We first provide experimental setup, followed by the exploration of the scale-invariance property for deep learning models. We then compare the results of the proposed methods with baseline methods in Section 4.3 and 4.4 on both normally trained models and adversarially trained models. Beyond the defense models based on adversarial training, we also quantify the effectiveness of the proposed methods on other advanced defense in Section 4.5. Additional discussions, the comparison between NI-FGSM and MI-FGSM and the comparison with classic attacks, are in Section 4.6. Code is available at https://github.com/JHL-HUST/SI-NI-FGSM.

---

**Algorithm 1** SI-NI-FGSM

---

**Input:** A clean example $x$ with ground-truth label $y^{true}$; a classifier $f$ with loss function $J$;
**Input:** Perturbation size $\epsilon$; maximum iterations $T$; number of scale copies $m$ and decay factor $\mu$.
**Output:** An adversarial example $x^{adv}$

1: $\alpha = \epsilon/T$
2: $g_0 = 0; x_0^{adv} = x$
3: **for** $t = 0$ to $T - 1$ **do**
4:     $g = 0$
5:     Get $x_t^{nes}$ by Eq.(6)      ▷ make a jump in the direction of previous accumulated gradients
6:     **for** $i = 0$ to $m - 1$ **do**     ▷ sum the gradients over the scale copies of the input image
7:         Get the gradients by $\nabla_x J(S_i(x_t^{nes}), y^{true})$
8:         Sum the gradients as $g = g + \nabla_x J(S_i(x_t^{nes}), y^{true})$
9:     Get average gradients as $g = \frac{1}{m} \cdot g$
10:     Update $g_{t+1}$ by $g_{t+1} = \mu \cdot g_t + \frac{g}{\|g\|_1}$
11:     Update $x_{t+1}^{adv}$ by Eq.(8)
12: **return** $x^{adv} = x_T^{adv}$

---

## 4.1 EXPERIMENTAL SETUP

**Dataset.** We randomly choose 1000 images belonging to the 1000 categories from ILSVRC 2012 validation set, which are almost correctly classified by all the testing models.

**Models.** For normally trained models, we consider Inception-v3 (Inc-v3) (Szegedy et al., 2016), Inception-v4 (Inc-v4), Inception-Resnet-v2 (IncRes-v2) (Szegedy et al., 2017) and Resnet-v2-101 (Res-101) (He et al., 2016). For adversarially trained models, we consider Inc-v3$_{ens3}$, Inc-v3$_{ens4}$ and IncRes-v2$_{ens}$ (Tramr et al., 2018).

Additionally, we include other advanced defense models: high-level representation guided denoiser (HGD) (Liao et al., 2018), random resizing and padding (R&P) (Xie et al., 2018), NIPS-r3[1], feature distillation (FD) (Liu et al., 2019), purifying perturbations via image compression model (Comdefend) (Jia et al., 2019) and randomized smoothing (RS) (Cohen et al., 2019).

**Baselines.** We integrate our methods with DIM (Xie et al., 2019), TIM, and TI-DIM (Dong et al., 2019), to show the performance improvement of SI-NI-FGSM over these baselines. Denote our SI-NI-FGSM integrated with other attacks as SI-NI-DIM, SI-NI-TIM, and SI-NI-TIM-DIM, respectively.

**Hyper-parameters.** For the hyper-parameters, we follow the settings in (Dong et al., 2018) with the maximum perturbation as $\epsilon = 16$, number of iteration $T = 10$, and step size $\alpha = 1.6$. For MI-FGSM, we adopt the default decay factor $\mu = 1.0$. For DIM, the transformation probability is set to 0.5. For TIM, we adopt the Gaussian kernel and the size of the kernel is set to $7 \times 7$. For our SI-NI-FGSM, the number of scale copies is set to $m = 5$.

## 4.2 SCALE-INVARIANT PROPERTY

To validate the scale-invariant property of deep neural networks, we randomly choose 1,000 original images from ImageNet dataset and keep the scale size in the range of $[0.1, 2.0]$ with a step size 0.1. Then we feed the scaled images into the testing models, including Inc-v3, Inc-v4, IncRes-2, and Res-101, to get the average loss over 1,000 images.

As shown in Figure 1, we can easily observe that the loss curves are smooth and stable when the scale size is in range $[0.1, 1.3]$. That is, the loss values are very similar for the original and scaled images. So we assume that the scale-invariant property of deep models is held within $[0.1, 1.3]$, and we leverage the scale-invariant property to optimize the adversarial perturbations over the scale copies of the input images.

---

[1]https://github.com/anlthms/nips-2017/tree/master/mmd

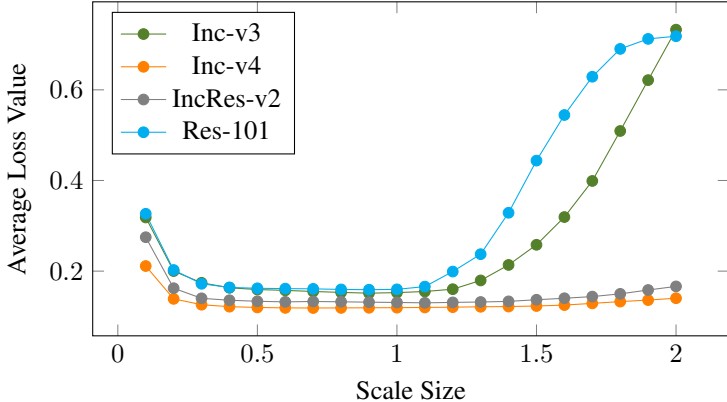

Figure 1: **The average losses for Inc-v3, Inc-v4, IncRes-v2 and Res-101 at each scale size.** The results are averaged over 1000 images.

### 4.3    ATTACKING A SINGLE MODEL

In this subsection, we integrate our SI-NI-FGSM with TIM, DIM and TI-DIM, respectively, and compare the black-box attack success rates of our extensions with the baselines under single model setting. As shown in Table 1, our extension methods consistently outperform the baseline attacks by $10\% \sim 35\%$ under the black-box setting, and achieve nearly 100% success rates under the white-box setting. It indicates that SI-NI-FGSM can serve as a powerful approach to improve the transferability of adversarial examples.

### 4.4    ATTACKING AN ENSEMBLE OF MODELS

Following the work of (Liu et al., 2016), we consider to show the performance of our methods by attacking multiple models simultaneously. Specifically, we attack an ensemble of normally trained models (including Inc-v3, Inc-v4, IncRes-v2 and Res-101) with equal ensemble weights using TIM, SI-NI-TIM, DIM, SI-NI-DIM, TI-DIM and SI-NI-TI-DIM, respectively.

As shown in Table 2, our methods improve the attack success rates across all experiments over the baselines. In general, our methods consistently outperform the baseline attacks by $10\% \sim 30\%$ under the black-box setting. Especially, SI-NI-TI-DIM, the extension by combining SI-NI-FGSM with TI-DIM, can fool the adversarially trained models with a high average success rate of 93.5%. It indicates that these advanced adversarially trained models provide little robustness guarantee under the black-box attack of SI-NI-TI-DIM.

### 4.5    ATTACKING OTHER ADVANCED DEFENSE MODELS

Besides normally trained models and adversarially trained models, we consider to quantify the effectiveness of our methods on other advanced defenses, including the top-3 defense solutions in the NIPS competition (high-level representation guided denoiser (HGD, rank-1) (Liao et al., 2018), random resizing and padding (R&P, rank-2) (Xie et al., 2018) and the rank-3 submission (NIPS-r3), and three recently proposed defense methods (feature distillation (FD) (Liu et al., 2019), purifying perturbations via image compression model (Comdefend) (Jia et al., 2019) and randomized smoothing (RS) (Cohen et al., 2019)).

We compare our SI-NI-TI-DIM with MI-FGSM (Dong et al., 2018), which is the top-1 attack solution in the NIPS 2017 competition, and TI-DIM (Dong et al., 2019), which is state-of-the-art attack. We first generate adversarial examples on the ensemble models, including Inc-v3, Inc-v4, IncRes-v2, and Res-101 by using MI-FGSM, TI-DIM, and SI-NI-TI-DIM, respectively. Then, we evaluate the adversarial examples by attacking these defenses.

As shown in Table 3, our method SI-NI-TI-DIM achieves an average attack success rate of 90.3%, surpassing state-of-the-art attacks by a large margin of 14.7%. By solely depending on the trans-

Table 1: **Attack success rates (%) of adversarial attacks against seven models under single-model setting.** The adversarial examples are crafted on Inc-v3, Inc-v4, IncRes-v2, and Res-101 respectively. * indicates the white-box attacks.

(a) **Comparison of TIM and the SI-NI-TIM extension.**

| Model | Attack | Inc-v3 | Inc-v4 | IncRes-v2 | Res-101 | Inc-v3$_{ens3}$ | Inc-v3$_{ens4}$ | IncRes-v2$_{ens}$ |
|---|---|---|---|---|---|---|---|---|
| Inc-v3 | TIM | **100.0*** | 47.8 | 42.8 | 39.5 | 24.0 | 21.4 | 12.9 |
| | SI-NI-TIM (**Ours**) | **100.0*** | **77.2** | **75.8** | **66.5** | **51.8** | **45.9** | **33.5** |
| Inc-v4 | TIM | 58.5 | 99.6* | 47.5 | 43.2 | 25.7 | 23.3 | 17.3 |
| | SI-NI-TIM (**Ours**) | **83.5** | **100.0*** | **76.6** | **68.9** | **57.8** | **54.3** | **42.9** |
| IncRes-v2 | TIM | 62.0 | 56.2 | 97.5* | 51.3 | 32.8 | 27.9 | 21.9 |
| | SI-NI-TIM (**Ours**) | **86.4** | **83.2** | **99.5*** | **77.2** | **66.1** | **60.2** | **57.1** |
| Res-101 | TIM | 59.0 | 53.6 | 51.8 | 99.3* | 36.8 | 32.2 | 23.5 |
| | SI-NI-TIM (**Ours**) | **78.3** | **74.1** | **73.0** | **99.8*** | **58.9** | **53.9** | **43.1** |

(b) **Comparison of DIM and the SI-NI-DIM extension.**

| Model | Attack | Inc-v3 | Inc-v4 | IncRes-v2 | Res-101 | Inc-v3$_{ens3}$ | Inc-v3$_{ens4}$ | IncRes-v2$_{ens}$ |
|---|---|---|---|---|---|---|---|---|
| Inc-v3 | DIM | 98.7* | 67.7 | 62.9 | 54.0 | 20.5 | 18.4 | 9.7 |
| | SI-NI-DIM (**Ours**) | **99.6*** | **84.7** | **81.7** | **75.4** | **36.9** | **34.6** | **20.2** |
| Inc-v4 | DIM | 70.7 | 98.0* | 63.2 | 55.9 | 21.9 | 22.3 | 11.9 |
| | SI-NI-DIM (**Ours**) | **89.7** | **99.3*** | **84.5** | **78.5** | **47.6** | **45.0** | **28.9** |
| IncRes-v2 | DIM | 69.1 | 63.9 | 93.6* | 57.4 | 29.4 | 24.0 | 17.3 |
| | SI-NI-DIM (**Ours**) | **89.7** | **86.4** | **99.1*** | **81.2** | **55.0** | **48.2** | **38.1** |
| Res-101 | DIM | 75.9 | 70.0 | 71.0 | 98.3* | 36.0 | 32.4 | 19.3 |
| | SI-NI-DIM (**Ours**) | **88.7** | **84.2** | **84.4** | **99.3*** | **52.4** | **48.0** | **33.2** |

(c) **Comparison of TI-DIM and the SI-NI-TI-DIM extension.**

| Model | Attack | Inc-v3 | Inc-v4 | IncRes-v2 | Res-101 | Inc-v3$_{ens3}$ | Inc-v3$_{ens4}$ | IncRes-v2$_{ens}$ |
|---|---|---|---|---|---|---|---|---|
| Inc-v3 | TI-DIM | 98.5* | 66.1 | 63.0 | 56.1 | 38.6 | 34.9 | 22.5 |
| | SI-NI-TI-DIM (**Ours**) | **99.6*** | **85.5** | **80.9** | **75.7** | **61.5** | **56.9** | **40.7** |
| Inc-v4 | TI-DIM | 72.5 | 97.8* | 63.4 | 54.5 | 38.1 | 35.2 | 25.3 |
| | SI-NI-TI-DIM (**Ours**) | **88.1** | **99.3*** | **83.7** | **77.0** | **65.0** | **63.1** | **49.4** |
| IncRes-v2 | TI-DIM | 73.2 | 67.5 | 92.4* | 61.3 | 46.4 | 40.2 | 35.8 |
| | SI-NI-TI-DIM (**Ours**) | **89.6** | **87.0** | **99.1*** | **83.9** | **74.0** | **67.9** | **63.7** |
| Res-101 | TI-DIM | 74.9 | 69.8 | 70.5 | 98.7* | 52.6 | 49.1 | 37.8 |
| | SI-NI-TI-DIM (**Ours**) | **86.4** | **82.6** | **84.6** | **99.0*** | **72.6** | **66.8** | **56.4** |

Table 2: **Attack success rates (%) of adversarial attacks against seven models under multi-model setting.** * indicates the white-box models being attacked.

| Attack | Inc-v3* | Inc-v4* | IncRes-v2* | Res-101* | Inc-v3$_{ens3}$ | Inc-v3$_{ens4}$ | IncRes-v2$_{ens}$ |
|---|---|---|---|---|---|---|---|
| TIM | 99.9 | 99.3 | 99.3 | 99.8 | 71.6 | 67.0 | 53.2 |
| SI-NI-TIM (**Ours**) | **100.0** | **100.0** | **100.0** | **100.0** | **93.2** | **90.1** | **84.5** |
| DIM | 99.7 | 99.2 | 98.9 | 98.9 | 66.4 | 60.9 | 41.6 |
| SI-NI-DIM (**Ours**) | **100.0** | **100.0** | **100.0** | **99.9** | **88.2** | **85.1** | **69.7** |
| TI-DIM | 99.6 | 98.8 | 98.8 | 98.9 | 85.2 | 80.2 | 73.3 |
| SI-NI-TI-DIM (**Ours**) | **99.9** | **99.9** | **99.9** | **99.9** | **96.0** | **94.3** | **90.3** |

ferability of adversarial examples and attacking on the normally trained models, SI-NI-TI-DIM can fool the adversarially trained models and other advanced defense mechanism, raising a new security issue for the development of more robust deep learning models. Some adversarial examples generated by SI-NI-TI-DIM are shown in Appendix B.

Table 3: **Attack success rates (%) of adversarial attacks against the advanced defense methods.**

| Attack | HGD | R&P | NIPS-r3 | FD | ComDefend | RS | **Average** |
|---|---|---|---|---|---|---|---|
| MI-FGSM | 36.9 | 29.3 | 40.8 | 51.6 | 47.5 | 27.1 | 38.9 |
| TI-DIM | 84.8 | 75.3 | 80.7 | 83.5 | 79.1 | 50.3 | 75.6 |
| SI-NI-TI-DIM (**Ours**) | **96.1** | **91.3** | **94.4** | **95.4** | **93.3** | **71.4** | **90.3** |

### 4.6 FURTHER ANALYSIS

**NI-FGSM vs. MI-FGSM.** We perform additional analysis for the difference between NI-FGSM with MI-FGSM (Dong et al., 2018). The adversarial examples are crafted on Inc-v3 with various number of iterations ranging from 4 to 16, and then transfer to attack Inc-v4 and IncRes-v2. As shown in Figure 2, NI-FGSM yields higher attack success rates than MI-FGSM with the same number of iterations. In another view, NI-FGSM needs fewer number of iterations to gain the same attack success rate of MI-FGSM. The results not only indicate that NI-FGSM has a better transferability, but also demonstrate that with the property of looking ahead, NI-FGSM can accelerate the generation of adversarial examples.

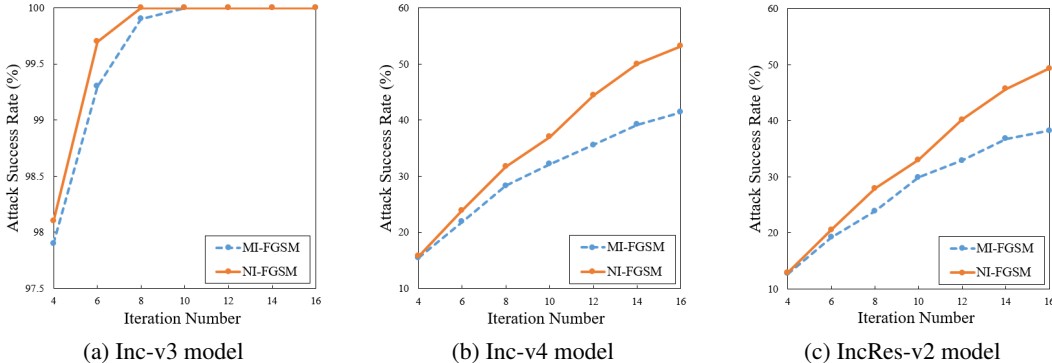

(a) Inc-v3 model  (b) Inc-v4 model  (c) IncRes-v2 model

Figure 2: **Attack success rates (%) of NI-FGSM and MI-FGSM on various number of iterations.** The adversarial examples are crafted on Inc-v3 model against (a) Inc-v3 model, (b) Inc-v4 model and (c) IncRes-v2 model.

**Comparison with classic attacks.** We consider to make addition comparison with classic attacks, including FGSM (Goodfellow et al., 2014), I-FGSM (Kurakin et al., 2016), PGD (Madry et al., 2018) and C&W (Carlini & Wagner, 2017). As shown in Table 4, our methods achieve 100% attack success rate which is the same as C&W under the white-box setting, and significantly outperform other methods under the black-box setting.

Table 4: **Attack success rates (%) of adversarial attacks against the models.** The adversarial examples are crafted on Inc-v3 using FGSM, I-FGSM, PGD, C&W, NI-FGSM, and SI-NI-FGSM. * indicates the white-box model being attacked.

| Attack | Inc-v3* | Inc-v4 | IncRes-v2 | Res-101 | Inc-v3$_{ens3}$ | Inc-v3$_{ens4}$ | IncRes-v2$_{ens}$ | **Average** |
|---|---|---|---|---|---|---|---|---|
| FGSM | 67.1 | 26.7 | 25.0 | 24.4 | 10.5 | 10.0 | 4.5 | 24.0 |
| I-FGSM | 99.9 | 20.7 | 18.5 | 15.3 | 3.6 | 5.8 | 2.9 | 23.8 |
| PGD | 99.5 | 17.3 | 15.1 | 13.1 | 6.1 | 5.6 | 3.1 | 20.9 |
| C&W | **100.0** | 18.4 | 16.2 | 14.3 | 3.8 | 4.7 | 2.7 | 22.9 |
| NI-FGSM (**Ours**) | **100.0** | 52.6 | 51.4 | 41.0 | 12.9 | 12.8 | 6.4 | 39.6 |
| SI-NI-FGSM (**Ours**) | **100.0** | **76.0** | **73.3** | **67.6** | **31.6** | **30.0** | **17.4** | **56.6** |

## 5 CONCLUSION AND FUTURE WORK

In this work, we propose two new attack methods, namely Nesterov Iterative Fast Gradient Sign Method (NI-FGSM) and Scale-Invariant attack Method (SIM), to improve the transferability of adversarial examples. NI-FGSM aims to adopt Nesterov accelerated gradient method into the gradient-based attack, and SIM aims to achieve *model augmentation* by leveraging the scale-invariant property of models. NI-FGSM and SIM can be naturally combined to build a robust attack, namely SI-NI-FGSM. Moreover, by integrating SI-NI-FGSM with the baseline attacks, we can further improve the transferability of adversarial examples. Extensive experiments demonstrate that our methods not only yield higher success rates on adversarially trained models but also break other strong defense mechanism.

Our work of NI-FGSM suggests that other momentum methods (e.g. Adam) may also be helpful to build a strong attack, which will be our future work, and the key is how to migrate the optimization method to the gradient-based iterative attack. Our work also shows that deep neural networks have the scale-invariant property, which we utilized to design the SIM to improve the attack transferability. However, it is not clear why the scale-invariant property holds. Possibly it is due to the batch normalization at each convolutional layer, that may mitigate the impact of the scale change. We will also explore the reason more thoroughly in our future work.

ACKNOWLEDGMENTS

This work is supported by the Fundamental Research Funds for the Central Universities (2019kfyXKJC021) and Microsoft Research Asia.

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

## A    DETAILS OF THE ALGORITHMS

The algorithm of SI-NI-TI-DIM attack is summarized in Algorithm 2. We can get the SI-NI-DIM attack algorithm by removing Step 10 of Algorithm 2, and get the SI-NI-TIM attack algorithm by removing $T(\cdot; p)$ in Step 7 of Algorithm 2.

---

**Algorithm 2** SI-NI-TI-DIM

---

**Input:** A clean example $x$ with ground-truth label $y^{true}$; a classifier $f$ with loss function $J$;
**Input:** Perturbation size $\epsilon$; maximum iterations $T$; number of scale copies $m$ and decay factor $\mu$.
**Output:** An adversarial example $x^{adv}$

1: $\alpha = \epsilon/T$
2: $g_0 = 0; x_0^{adv} = x$
3: **for** $t = 0$ to $T - 1$ **do**
4:     $g = 0$
5:     Get $x_t^{nes}$ by Eq.(6)        ▷ make a jump in the direction of previous accumulated gradients
6:     **for** $i = 0$ to $m - 1$ **do**        ▷ sum the gradients over the scale copies of the input image
7:         Get the gradients by $\nabla_x J(T(S_i(x_t^{nes}); p), y^{true})$ ▷ apply random resizing and padding to the inputs with the probability $p$
8:         Sum the gradients as $g = g + \nabla_x J(T(S_i(x_t^{nes}); p), y^{true})$
9:     Get average gradients as $g = \frac{1}{m} \cdot g$
10:     Convolve the gradients by $g = \boldsymbol{W} * g$ ▷ convolve gradient with the pre-defined kernel $\boldsymbol{W}$
11:     Update $g_{t+1}$ by $g_{t+1} = \mu \cdot g_t + \frac{g}{\|g\|_1}$
12:     Update $x_{t+1}^{adv}$ by Eq.(8)
13: **return** $x^{adv} = x_T^{adv}$

---

## B    VISUALIZATION OF ADVERSARIAL EXAMPLES

We visualize 12 randomly selected benign images and their corresponding adversarial images in Figure 3. The adversarial images are crafted on the ensemble models, including Inc-v3, Inc-v4, IncRes-v2 and Res-101, using the proposed SI-NI-TI-DIM. We see that these generated adversarial perturbations are human imperceptible.

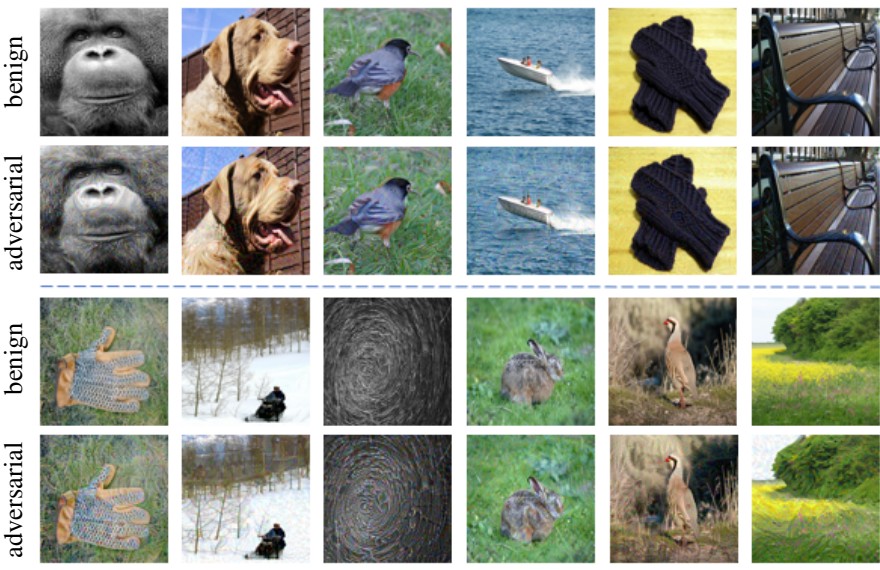

Figure 3: Visualization of randomly picked benign images and their corresponding adversarial images, crafted on the ensemble models using SI-NI-TI-DIM.

