# OpenReview forum: "Nesterov Accelerated Gradient and Scale Invariance for Adversarial Attacks"
_ICLR.cc/2020/Conference — Accept (Poster)_

### Official Review · AnonReviewer2 · 2019-10-20
**Official Blind Review #2**

**Rating:** 6

**Review:**

This paper studies how to generate transferable adversarial examples for black-box attacks. Two methods have been proposed,  namely Nesterov Iterative Fast Gradient Sign Method (NI-FGSM) and Scale-Invariant attack Method (SIM). The first method adopts Nesterov optimizer instead of momentum optimizer to generate adversarial examples. And the second is a model-augmentation method to avoid "overfitting" of the adversarial examples. Experiments on ImageNet can prove the effectiveness of the proposed methods.

Overall, this paper is well-written. The motivation of the proposed methods are generally clear although I have some questions. The experiments can generally prove the effectiveness.

My detailed questions about this paper are:
1. The motivation in Section 3.1, which regards generating adversarial examples as training models, and transferability as generalizability, is first introduced in Dong et al. (2018). The authors should acknowledge and refer to the previous work to present the motivation.
2. It's not clear why deep neural networks have the scale-invariant property. Is it due to that a batch normalization layer is usually applied after the first conv layer to mitigate the effect of scale change?
3. It's not fair to directly compare DIM with SI-NI-DIM (also TIM vs. SI-NI-TIM; TI-DIM vs. SI-NI-TI-DIM), since SI-NI needs to calculate the gradient over 5 ensembles. It's better to compare the performance of two methods with the same number of gradient calculations.
4. Is there an efficient way of calculating the gradient for scale-invariant attacks like translation-invariant attacks in Dong et al. (2019)?

**Experience Assessment:**

I have published in this field for several years.

**Review Assessment: Checking Correctness Of Derivations And Theory:**

N/A

**Review Assessment: Checking Correctness Of Experiments:**

I carefully checked the experiments.

**Review Assessment: Thoroughness In Paper Reading:**

I read the paper thoroughly.

---

> ### Author Response · Authors · 2019-11-14
> **RE: Review #2**
>
> Thank you for your insightful comments. We have performed the corresponding revisions based on your constructive suggestions.
>
> A1. Thanks for reminding us. Following your suggestion, we have updated in Sec. 3.1 and cited Dong et al. (2018) in the revision.
>
> A2. Thanks for your insightful and valuable suggestion. Yes, batch normalization may mitigate the impact of the scale change. If we consider a layer $t$ with a ReLu activation function $g(\cdot)$, followed by a batch normalization layer $BN(\cdot): z = BN(RL(Wx^t+b))$, where $x^t$ is the current layer’s input, the weight matrix $W$ (Convolution operator can also be regarded as a sparse linear operation $W$) and bias vector $b$ are the current layer’s parameters.
> When at layer 0, we scale an input $x$ with factor $s$, we can get:
> $z = BN(g(W(s \cdot x)+b)) = BN(s \cdot g[(Wx+b/s)] ).$
> Therefore, for various scales on the input $x$, we have similar outputs with slightly different truncations by ReLu, and BN will shrink their difference and mitigate the impact of the scale change.
> Similarly, at the next layer after ReLu and BN, the differences will be shrunk again. After several Conv+ReLu+BN layers, the output will have some scale invariance for a range of the scales on the input.
>
> In Figure 1, Inc-v3 and Inc-v4, Inc-v4 is deeper than Inc-v3, and shows a better scale invariance, which is consistent with the above explanation. For Res-101 and IncRes-v2, their scale ranges are smaller due to the residual connection among the connections, which is also consistent with the above explanation. Due to the time limit, we could not try more experiments to demonstrate this observation, but we are running the experiments, and will add in the final version.
>
> A3. In this paper, we consider to demonstrate that our method SI-NI is easy to combine with other orthogonal state-of-the-art methods to boost the transferability of adversarial examples. Following this spirit, we directly compare SI-NI-DIM with DIM (also TIM vs. SI-NI-TIM; TI-DIM vs. SI-NI-TI-DIM). Under such motivation, it is fair to conduct the experiments with the same number of iterations to show the improvement of the performance under the same parameters.
>
> Following your suggestion, we also did a comparison on the performance with the same number of gradient calculations.  We increase the number of iterations to 50 for TIM, DIM and TI-DIM, denote as TIM (50), DIM (50) and TI-DIM (50), respectively. The results are as follows:
>
> 1) For TIM, TIM (50) gets the worse performance than TIM (10).
> 2) For DIM, DIM (50) improves the transferability for normally trained models but reduces the transferability for adversarially trained models.
> 3) The case of TI-DIM is the same as that of DIM.
>
> Overall, as comparing the performance with the same number of gradient calculations, our method still achieves a considerable improvement on the transferability for normally trained models and adversarially trained models.
>
> ---------------------------------------------------------------------------------------------------------------------------
> | Attack  | Inc-v3* | Inc-v4 | IncRes-v2 | Res-101 | Inc-v3$_{ens3}$ | Inc-v3$_{ens4}$ | IncRes-v2$_{ens}$ |
> ---------------------------------------------------------------------------------------------------------------------------
> | TIM (10)                | 100  | 47.8 | 42.8 | 39.5 | 24.0 | 21.4 | 12.9 |
> | TIM (50)                | 100  | 46.9 | 41.4 | 36.8 | 15.3 | 14.7 |  8.2  |
> | SI-NI-TIM (10)      | 100  | 77.2 | 75.8 | 66.5 | 51.8 | 45.9 | 33.5 |
> | DIM (10)                | 98.7 | 67.7 | 62.9 | 54.0 | 20.5 | 18.4 |  9.7  |
> | DIM (50)                | 99.1 | 77.2 | 71.9 | 61.7 | 13.1 | 13.8 |  6.1  |
> | SI-NI-DIM (10)      | 99.6 | 84.7 | 81.7 | 75.4 | 36.9 | 34.6 | 20.2 |
> | TI-DIM (10)           | 98.7 | 66.1 | 63.0 | 56.1 | 38.6 | 34.9 |  22.5 |
> | TI-DIM (50)           | 100  | 78.2 | 71.7 | 63.3 | 27.6 | 25.4 |  15.6 |
> | SI-NI-TI-DIM (10) | 99.6 | 85.5 | 80.9 | 75.7 | 61.5 | 56.9 |  40.7 |
> ------------------------------------------------------------------------------------------
> (Note: the adversarial examples are crafted for Inc-v3, which means it is white-box attack for Inc-v3 and black-box attack for other models)
>
> A4. Thank you for your suggestion. But we are afraid there is no efficient way to speed up the gradient calculation for scale-invariant attacks.
> 1)	For the work of Dong et al. (2019), they make an assumption that the translation-invariant property is nearly held with very small translations. Based on this assumption, it is easy to get the gradient of the translated image
>       $T_{ij}(x)$ by translating the gradient of the original image $x$.
> 2)	But in our scale-invariant method, as for one layer of conv+relu+batch-norm,
>       $z = BN(g(W(s \cdot x)+b)) = BN(s \cdot g[(Wx+b/s)] ).$
> where $W$ is a sparse matrix representing the conv operator. The gradient of $s \cdot x$ could not be calculated by scaling the gradient of $x$ with $s$.

---

### Official Review · AnonReviewer3 · 2019-10-23
**Official Blind Review #3**

**Rating:** 6

**Review:**

In this paper, the authors apply the Nesterov Accelerated Gradient method to the adversarial attack task and achieve better transferability of the adversarial examples. Furthermore, the authors introduce a scale transformation method to provide the augmentation on the model, which also boosts the transferability of the attack method. Experiments are carried out to verify the scale-invariant property and the Nesterov Accelerated Gradient method on both single and ensemble of models. All experiments turn out to be a positive support to the authors' claim.

However, one small drawback of this paper is that the author does not claim any comparison between the Nesterov Accelerated Gradient Method and other momentum methods (e.g. Adam, momentum-SGD, etc). This experiment is somehow important since it shows the better transformability is obtained from 1) Nesterov Accelerated Gradient Method only, or 2) all momentum method, which is significant for further research.

Also, in the setting of the Scale-Invariant Transformation, the authors forget to address that what if the attacked network has an input normalization. Does it mean to downsample the value of each pixel in the input image? If so, is the equation $S_i(x) = x / 2^i$ better to be $S_i(x) = [x / 2^i]$ where $[]$ means casting to the nearest integer?

One more question of this work is:  The Nesterov Accelerated Gradient method is known for its proveable fast descent property comparing to the traditional Gradient method. Do you observe any speed-up during your training?

**Experience Assessment:**

I have read many papers in this area.

**Review Assessment: Checking Correctness Of Derivations And Theory:**

I carefully checked the derivations and theory.

**Review Assessment: Checking Correctness Of Experiments:**

I assessed the sensibility of the experiments.

**Review Assessment: Thoroughness In Paper Reading:**

I read the paper thoroughly.

---

> ### Author Response · Authors · 2019-11-14
> **RE: Review #3**
>
> Thank you for your insightful comments. We have performed the corresponding revision based on your constructive suggestions.
>
> A1. Thank you for the valuable suggestion. We agree that the comparison of Nesterov Accelerated Gradient (NAG) with other momentum methods is important. Indeed, we have compared our NI-FGSM with MI-FGSM (Momentum Iterative Fast Gradient Sign Method), and in the revision, we make a more thorough comparison with various number of iterations (Figure 2) and provide more analysis in Sec. 4.6.
> For other momentum methods (e.g. Adam, momentum-SGD, etc), we also think it is a good direction to try for adversarial attacks, and the key is how to migrate the optimization method to the gradient-based iterative attack. We add the description in conclusion as our future work. Thank you so much.
>
> A2. Our scale operation is conducted after the input normalization. For example for inception_v3, the input normalization will scale the input image $x$ to $[-1, 1]$, and we will then apply scale operation so the range of input after SIM is $[-1/s, 1/s]$. The pixel value is in float type, so there is no need to downsample.
>
> A3. Thank you for the insightful comments. We perform additional analysis for the difference between NI-FGSM with MI-FGSM (Figure 2 in the revision). The adversarial examples are crafted on Inc-v3 with various number of iterations ranging from 4 to 16, and then transfer to attack Inc-v4 and IncRes-v2.  As shown in Figure 2, NI-FGSM yields higher attack success rates than MI-FGSM with the same number of iterations. In another view, NI-FGSM needs fewer number of iterations to gain the same attack success rate of MI-FGSM. The results not only indicate that NI-FGSM has a better transferability, but also demonstrate that with the property of looking ahead, NI-FGSM can accelerate the generation of adversarial examples.

---

### Official Review · AnonReviewer1 · 2019-10-27
**Official Blind Review #1**

**Rating:** 3

**Review:**

In this paper, the authors proposed two methods of Nesterov Iterative Fast Gradient Sign Method (NI-FGSM) and Scale-Invariant attack Method (SIM) to improve the transferability of adversarial examples. Empirical results on ImageNet dataset demonstrate its effectiveness. In general, the paper is clearly written and easy to follow but I still have several concerns:
1.	Although the method is easy to understand, the authors are expected to clarify why the methods can improve the transferability. The authors are expected to make more theoretical analysis.
2.	The authors are expected to make more comprehensive comparisons with the recent methods in adversarial attacks, e.g, PGD, and C&W even if some methods are designed for white-box attack.
3.	The authors are expected to make more evaluations on the models with defense mechanism, and numerous important methods are missing. Without this, the authors cannot claim its effectiveness since only experiments on NIPS2017 is not enough.


**Experience Assessment:**

I have published in this field for several years.

**Review Assessment: Checking Correctness Of Derivations And Theory:**

I carefully checked the derivations and theory.

**Review Assessment: Checking Correctness Of Experiments:**

I assessed the sensibility of the experiments.

**Review Assessment: Thoroughness In Paper Reading:**

I read the paper thoroughly.

---

> ### Author Response · Authors · 2019-11-14
> **RE: Review #1**
>
> Thank you for your insightful comments. We have performed the corresponding revision based on your constructive suggestions.
>
> A1. We acknowledge that a solid theoretical analysis on why our attack methods can improve the transferability is very important. However, though many efficient adversarial attack methods have been proposed in the literature, so far there is little theoretical results on the transferability, and researchers usually provide some intuitive explanation or just provide some empirical evidence. We will try our best to answer your question as follows:
>
> (1) For NI-FGSM: Typical gradient-based iterative attacks (e.g., I-FGSM) greedily perturb the images in the direction of the sign of the gradient at each iteration, which usually falls into poor local maxima, and shows weak transferability than single-step attacks (e.g., FGSM). Dong et al. [1] show that adopting momentum into attacks can stabilize the update directions, which helps to escape from poor local maxima and improve the transferability. Compared to momentum, beyond stabilize the update directions, the anticipatory update of NAG gives previous accumulated gradient a correction that helps to effectively look ahead. Such looking ahead property of NAG can help us escape from poor local maxima easier and faster, resulting in the improvement on transferability.
>
> To provide empirical evidence, we conduct additional experiments in Figure 2 in the revision. It shows that, using the same number of iterations, NI-FGSM yields higher attack success rates than MI-FGSM (Momentum Iterative Fast Gradient Sign Method), not only for the white-box setting (a), but even better for the black-box settings (b and c), demonstrating that NI-FGSM has a better transferability.
>
> (2) For SIM: Similarly to the generalization of models can be improved by feeding more training data, the transferability of adversarial examples can also be improved by attacking more models simultaneously [1, 2]. Essentially, SIM derives an ensemble of models to be attacked from the original model via the loss-preserving scale transformation. Such model argumentation will help improve the transferability of adversarial examples.
>
> In the revision, we also add more explanation in Sec. 3.2.
>
> A2. Following your constructive suggestion, we have made more comprehensive comparisons with some recent classic methods in adversarial attacks, including FGSM, I-FGSM, PGD and C&W. The results are reported in Table 4 in the revision (also list in the following). Under the white-box setting, our methods achieve 100% attack success rate, which is as good as C&W, and is better than FGSM, I-FGSM, and PGD. Under the black-box setting, our methods significantly outperform all the four baseline methods.
>
> ---------------------------------------------------------------------------------------------------------------------------------------------
> Attack      |Inc-v3 |Inc-v4| IncRes-v2| Res-101 | Inc-v3$_{ens3}$ | Inc-v3$_{ens4}$ | IncRes-v2$_{ens}$ | AVG|
> --------------------------------------------------------------------------------------------------------------------------------------------
> FGSM            |   67.1  |   26.7  |     25.0    |   24.4    |    10.5   |    10.0   |     4.5    |  24.0
> I-FGSM         |   99.9  |   20.7  |     18.5    |   15.3    |     3.6    |     5.8     |     2.9    |  23.8
> PGD              |   99.5  |   17.3  |     15.1    |   13.1    |     6.1    |     5.6     |     3.1    |  20.9
> C&W             | 100.0  |   18.4  |     16.2    |   14.3    |     3.8    |     4.7     |    2.7     |  22.9
> NI-FGSM      | 100.0  |   52.6  |     51.4    |   41.0    |    12.9   |    12.8    |    6.4     |  39.6
> SI-NI-FGSM | 100.0  |   76.0  |     73.3    |   67.6    |    31.6   |    30.0    |   17.4    |  56.6
> ---------------------------------------------------------------------------------------------------------------------------------------------
>
> A3. In the revision, we evaluate the attack performance in Table 3 with three more advanced defense methods, which have shown to be robust on the ImageNet dataset, namely FD (Feature Distillation) [3], ComDefend [4], and RS (Randomized Smoothing) [5]. Our method SI-NI-TI-DIM achieves an average attack success rate of 90.3% on the six advanced defense methods, surpassing the state-of-the-art method TI-DIM [6] by a large margin of 14.7%. Thus we can claim the effectiveness of our methods. Thank you for the valuable suggestion.
>
> [1] Boosting adversarial attacks with momentum. CVPR 2018
> [2] Delving into Transferable Adversarial Examples and Black-box Attacks. ICLR 2017
> [3] Feature Distillation: DNN-Oriented JPEG Compression Against Adversarial Examples. CVPR2019
> [4] ComDefend: An Efficient Image Compression Model to Defend Adversarial Examples. CVPR2019
> [5] Certified Adversarial Robustness via Randomized Smoothing. ICML 2019
> [6] Evading defenses to transferable adversarial examples by translation-invariant attacks. CVPR 2019

---

### Author Response · Authors · 2019-11-14
**General Response**

We deeply appreciate all reviewers for the thorough comments and valuable suggestions, which definitely help the improvement of our paper.

We would like to briefly summarize our modification in the updated paper and provide specific response in the individual comment.

Our main modifications are as follows:

- We have polished the overall writing.

- In Section 2.3, we introduce and discuss more advanced attack and defense methods.

- In Section 3.2, we add more explanation on why our NI-FGSM method can improve the transferability.

- In Section 4.3, we rewrite this subsection and add an experiment to further show the advantage of SI-NI-FGSM.

- In Section 4.5, we evaluate our methods on six most recent defense techniques to further demonstrate the effectiveness of our methods.

- In Section 4.6, we made further analysis to compare NI-FGSM and MI-FGSM, and also do comparison with several popular adversarial attacks (FGSM, I-FGSM, PGD and C&W).

- In Section 5, the conclusion section, we add one paragraph for our future work on NI-FGSM and SIM.

By regarding the process of generating adversarial examples as an optimization problem, we propose two novel attack methods to improve the transferability of adversarial examples. We provide intuitive explanations as well as strong and extensive empirical support for the effectiveness of the proposed methods.We believe our attack methods can serve as strong baselines to assess the robustness of deep learning algorithm.

We hope all our effort can make our paper more comprehensive and address most of your concerns. Thank you very much!

Best regards,
Authors

---

### Decision · Program_Chairs · 2019-12-19

**Decision:**

Accept (Poster)

**Comment:**

Under the optimization formulation of adversarial attack, this paper proposes two methods to improve the transferability of adversarial examples, namely Nesterov Iterative Fast Gradient Sign Method (NI-FGSM) and Scale-Invariant attack Method (SIM). NI-FGSM adapts Nesterov accelerated gradient into the iterative attacks to effectively look ahead and avoid the “missing” of the global maximum, and SIM optimizes the adversarial perturbations over the scale copies of the input images so as to avoid “overfitting” on the white-box model being attacked and generate more transferable adversarial examples. Empirical results demonstrate the effectiveness of the proposed methods. The ideas are sensible, and the empirical studies were strengthened during rebuttal.